# Effect of Samryungbaekchul-san Combined with Otilonium Bromide on Diarrhea-Predominant Irritable Bowel Syndrome: A Pilot Randomized Controlled Trial

**DOI:** 10.3390/jcm8101558

**Published:** 2019-09-27

**Authors:** Jin-Hyun Lee, Joong Il Kim, Myong Ki Baeg, Yun-Young Sunwoo, Kwangsun Do, Jung-Han Lee, Hye-Jung Kim, Ja Sung Choi, Jayoung Kim, Chang-Seob Seo, Hyeun-Kyoo Shin, Hyekyung Ha, Tae-Yong Park

**Affiliations:** 1Institute for Integrative Medicine, Catholic Kwandong University International St. Mary’s Hospital, Incheon 22711, Korea; doolyjinhyun@empal.com (J.-H.L.); baegmk@gmail.com (M.K.B.); 545544@ish.ac.kr (K.D.); gkimmhj@ish.ac.kr (H.-J.K.); cjs0123@ish.ac.kr (J.S.C.); lmkjy7@gmail.com (J.K.); 2Future Medicine Division, Korea Institute of Oriental Medicine, Daejeon 34054, Korea; jikim@kiom.re.kr; 3Iksoodang Korean Medical Clinic, Incheon 21425, Korea; 127501@hanmail.net; 4Department of Korean Rehabilitation Medicine, Wonkwang University College of Korean Medicine, Iksan 54383, Korea; milpaso75@daum.net; 5Herbal Medicine Research Division, Korea Institute of Oriental Medicine, Daejeon 34054, Korea; csseo0914@kiom.re.kr (C.-S.S.); hkshin@kiom.re.kr (H.-K.S.)

**Keywords:** irritable bowel syndrome, functional bowel disorder, herbal extract, complementary medicine

## Abstract

Conventional and herbal drugs are frequently used together to treat many disorders. Samryungbaekchul-san (SRS, a herbal formula) and otilonium bromide (OB, an antispasmodic agent) are widely used to treat diarrhea-predominant irritable bowel syndrome (D-IBS) in Eastern Asian countries. However, there have been no studies on the co-administration of SRS and OB. Therefore, we aimed to preliminarily assess the feasibility of SRS combined with OB for D-IBS treatment in a pilot double-blind, four-arm, parallel-group, randomized controlled trial (RCT), including 80 patients diagnosed with D-IBS according to the Rome III criteria. The patients were randomly assigned to four treatment groups and were administered drugs for eight weeks after a two-week preparatory period. Follow-up was conducted four weeks after the administration period. The primary outcome was evaluated by using a global D-IBS symptom improvement score; no statistically significant difference was observed between the groups. However, multiple logistic regression analysis of primary outcome scores shows that SRS significantly improved D-IBS symptoms (*p* < 0.05). For secondary outcomes, better results were observed in the SRS + OB group, in terms of symptoms, including abdominal pain, discomfort, frequency of abdominal pain, and stool form than in OB alone or placebo groups (*p* < 0.05). In conclusion, the co-administration of SRS and OB might be an effective and safe strategy for the treatment of D-IBS. Large-scale RCTs are warranted to further confirm and clarify these findings.

## 1. Introduction

Irritable bowel syndrome (IBS) is a very common disorder worldwide, and is associated with a high prevalence (10–15%), especially in the West (Europe and North America) [1,2,3]. IBS has a negative impact on the quality of life (QOL) and the psychological state of the patient [4]. The disease increases the economic burden on employers as it lowers labor and social capacities of the affected individuals [5,6,7,8]. In general, IBS is a chronic functional gastrointestinal (GI) disorder characterized by recurrent abdominal pain and abnormal bowel habits without structural and biochemical abnormalities [9,10,11]. The Rome IV criteria (2016) classify IBS into three types based on bowel pattern symptoms as follows: Diarrhea-predominant IBS (D-IBS), constipation-predominant IBS, and mixed IBS [10].

The drugs most commonly used to treat IBS are smooth muscle relaxants, bulking agents, antidiarrheal agents, and pain management drugs [12,13]. However, these conventional treatments have certain limitations. Therefore, attempts have been made to understand IBS from the perspectives of psychosocial stress and the enteric nervous system [14], and accordingly, various alternative methods have been used for its treatment including dietary regulation, psychotherapy, acupuncture, and herbal drugs [15,16,17,18]. Samryungbaekchul-san (SRS, Jinryobyakujutsu-san in Kampo medicine and Shenlingbaizhu-san in traditional Chinese medicine) is a herbal formula that has been used since the Song Dynasty (AD 1107) in China. SRS has anti-barium chloride and anti-acetylcholine effects that lead to enhanced immune functions, protein expression related to intestinal absorption, and the suppression of gastric secretions and small intestinal motility [19,20]. In traditional Chinese and Korean medicine, it is widely used for diarrhea and abdominal pain [21,22,23,24].

Otilonium bromide (OB) is used to alleviate symptoms of D-IBS [25,26]. In pre-clinical trials, it was found to inhibit L-type and T-type calcium channels, muscarinic receptors, human sigmoid motility, and tachykininergic responses [27,28,29]. In clinical studies, OB was also shown to improve IBS symptoms [30,31,32]. The World Gastroenterology Organization recommends the use of antispasmodic agents for IBS [3]. Although herbal and conventional drugs are widely used in D-IBS in Eastern Asian countries, there has been no investigation of the efficacy and safety of SRS and OB co-administration, and it is, thus, difficult to apply this strategy in clinical practice based on the lack of available evidence.

We hypothesized that because the main effects of these two drugs are different, the co-administration of OB and SRS would have synergistic effects that would improve D-IBS symptoms, diarrhea-like stool patterns (the main effect of SRS is antidiarrheic) [19,33], and abdominal discomfort or pain (the main effect of OB is antispasmodic activity) [30]. Based on these previous studies and our hypothesis, we aimed to assess the feasibility of SRS and OB co-administration, and acquire information on a suitable design to conduct further large-scale clinical trials as the current study is a preliminary trial.

## 2. Methods

### 2.1. Ethics and Trial Registration

We obtained ethical approval from the ethics committee of Catholic Kwandong University International St. Mary’s Hospital (IS15MISV0033) and the Korean Ministry of Food and Drug Safety (KMFDS, 30769, clinical trial registration No. KCT0001621, version 4.0, http://cris.nih.go.kr). The study protocol was published in the BMJ Open Journal in December 2017 [34].

### 2.2. Study Design

The study planning and design began in April 2015. This was a pilot double-blinded placebo-controlled, double-dummy parallel RCT, including 80 patients diagnosed with D-IBS at the Catholic Kwandong University International St. Mary’s Hospital, Incheon, South Korea. Participants were recruited through advertisements at internet sites, the hospital, and the subway from March 2016 to December 2017. According to the eligibility criteria (Table 1), ineligible participants were excluded from the study. All participants provided written informed consent after providing them with a sufficient explanation of the study.

The participants were randomly assigned to one of four treatment groups at a 1:1:1:1 ratio as follows: SRS + OB; SRS + placebo-OB (P-OB); placebo-SRS (P-SRS) + OB; and P-SRS + P-OB. Randomization was carried out by a statistician who did not have a role in the intervention administered to participants. A block randomization method was used, with block = 10 and size =8, using SAS 9.4 software (SAS Institute, Cary, NC, USA). The allocation was concealed by the researcher who performed the randomization. A randomization table was used to label the medications with the assignment numbers, and the drugs were administered by the managing pharmacist who was not involved in the data analysis.

The patients were required to participate in a preparatory two-week run-in period, during which screening tests were conducted (weeks −2 to 0). After this preparatory period, patients were given the drugs for eight weeks (weeks 0 to 8). During the treatment period, all patients and researchers involved in the trial were blinded, except for the researcher who had carried out randomization. Allocation concealment was ensured during medication administration. The patients were provided with drugs in opaque envelopes containing the assignment numbers. Post-treatment follow-up was conducted four weeks after the administration period (week 12). The study subjects were required to attend an appointment to evaluate outcomes and adverse events, including four in-person visits (at 0, 4, 8, and 12 weeks) and two scheduled telephone contacts (2 and 6 weeks). Baseline anthropometric measurements, body mass index (BMI), and relevant history were also recorded.

### 2.3. Sample Size Calculation

Similar clinical RCTs on IBS included 60–64 patients in total, and each arm consisted of 12–30 participants [35,36,37]. Based on these previous trials, the minimal clinical significance would be achieved when only 16 patients completed the study in each group. As we expected a 20% dropout rate, in this study, we reasoned that each group should comprise of at least 20 patients. Thus, the total sample size for minimal clinical significance was calculated as 80 participants.

### 2.4. Drug Formulation, Dosage, and Administration

SRS (SRS granule; Han Kook Shin Yak, Nonsan, Korea) consisted of 10 herbs, which were extracted with water and mixed with lactose and corn-starch in accordance with the Korean Good Manufacturing Practice (K-GMP, Table 2). The manufacture of SRS was regulated and allowed by the KMFDS. P-SRS was prepared with cornstarch, lactose, caramel, and powder such that it had a scent similar to that of Ginseng radix and shape, taste, color, and odor that were similar to those of SRS. These herbal drugs were sealed in identical aluminum bags with the same labeling and were provided to the patients in this form. The patients took the drugs after dissolving them in warm water 30 min before each meal (5 g/pack, three times per day) for eight weeks.

Phytochemicals in SRS and P-SRS were analyzed using a high-performance liquid chromatography-photodiode array (HPLC-PDA) method. Two optimized HPLC-PDA methods were used for the quantitative determination of three compounds (liquiritin, liquiritigenin, and glycyrrhizin) and one compound (allantoin) in placebo and SRS samples. None of the marker compounds was detected in the placebo sample. However, all four marker compounds were detected in the SRS sample based on comparisons of the retention times and UV spectra of reference standards. In the SRS sample, allantoin, liquiritin, liquiritigenin, and glycyrrhizin were detected at 6.55, 14.80, 21.10, and 26.94 min, respectively, and a concentration of 0.09, 0.08, 0.01, and 1.56 mg/g, respectively (Figure 1 and Figure 2).

OB (Menoctyl Tab; Dong Hwa Pharm, Seoul, Korea) was formulated as a tablet (40 mg). P-OB was prepared from Ludipress (lactose 94%, povidone 3%, and crospovidone 3%), microcrystalline cellulose, magnesium stearate, and Opadry II White (polyvinyl alcohol, talc, titanium oxide, polyethylene glycol). It had a size, taste, shape, and color that was similar to that of OB. The patients took one OB P-OB tablet 30 min before each meal (three times/day) over the 8-week administration period.

### 2.5. Study Outcomes

#### 2.5.1. Primary Outcome

The primary outcome (global D-IBS symptom improvement score) was assessed based on five patient responses (at 2 and 6 weeks via telephone and 4, 8, and 12 weeks during in-person visits), using the Subject’s Global Assessment of Relief (“How much do you think the symptoms of D-IBS have improved compared to before the clinical trial?”), which evaluates subjective symptom improvements in patients [38].

#### 2.5.2. Secondary Outcomes

As secondary outcomes, the severity of D-IBS symptoms was evaluated using a Likert scale as follows: (1) Abdominal pain, (2) abdominal discomfort, (3) satisfaction of defecation, (4) frequency of abdominal pain, and (5) QOL. The patients recorded the number of defecations per day, and were evaluated based on the Bristol stool form (BSF) scale, and the degree of force used in bowel movements at 0, 2, 4, 6, 8, and 12 weeks [39]. The severity of D-IBS symptoms was assessed at 2, 4, 6, 8, and 12 weeks from the initiation of treatment.

#### 2.5.3. Blood Chemistry and Immunologic Tests

Based on previous findings that IBS is related to stress and an intestinal inflammatory response [9,14,40], the following parameters were tested to investigate the mechanisms related to these reactions, at 0, 4, and 8 weeks: (1) Cortisol, (2) corticotropin-releasing hormone (CRH), (3) serotonin, and (4) cytokines, including eotaxin, FGF, G-CSF, GM-CSF, IFN-γ, IL-10, IL-12, IL-13, IL-15, IL-17, IL-1β, IL-1ra, IL-2, IL-4, IL-5, IL-6, IL-7, IL-8, IL-9, IP-10, MCP-1, MIP-1, PDGF, RANTES, TNF-α, and VEGF [41,42]. Enzyme-linked immunosorbent assay (ELISA) kits for cortisol and serotonin were purchased from LDN GmBH and Co. (Nordhorn, Germany), and a CRH ELISA kit was obtained from MyBioSource (San Diego, CA, USA). The levels of 27 cytokines were determined using the Bio-Plex Pro™ Human Cytokine 27-Plex (Bio-Rad Laboratories, Hercules, CA, USA).

#### 2.5.4. Adverse Event Monitoring

Adverse events were monitored during the trial through safety assessments and laboratory tests. All adverse reactions that occurred throughout the intervention and follow-up periods were evaluated according to the World Health Organization adverse reaction terminology [43], and were graded on a scale of 1 to 5 according to the Common Terminology Criteria for Adverse Events (version 4.03). The categories of expected adverse events were defined based on applicable product information or the characteristics of SRS and OB. For adverse effects associated with concomitant drugs, the duration of administration, dosage, composition, and an indication of those drugs were recorded in detail. Blood and urine tests were conducted to determine adverse reactions.

#### 2.5.5. Measurement of Compliance and Blinding

Compliance was determined by counting returned SRS or OB. To measure the success of blinding, participants were asked whether the drug they had been taking was either real or a placebo at the end of the trial (week 8).

### 2.6. Statistical Analyses

All statistical analyses were performed by blinded statisticians using SAS version 9.1.3 (SAS Institute, Cary, NC, USA). All data are reported as the mean ± standard error of the mean, and a *p* < 0.05 was considered statistically significant. To assess the efficacy of the trial, both intent-to-treat (ITT) analysis (main analytical method; especially validity analysis) and per-protocol (PP) analysis (secondary analytical method) were performed. To compare continuous variables between the groups with respect to baseline characteristics and primary outcome, one-way analysis of variance (ANOVA), Kruskal-Wallis or Chi-square tests were conducted for each assessment point. The incidence of adverse events and abnormal experimental results were reported according to the groups, and relevant non-parametric tests were performed (a paired *t*-test for continuous data, McNemar’s test for categorical data, and Fisher’s exact test for adverse events). The CATMOD procedure was applied to evaluate the rate of symptom relief. A paired *t*-test, Wilcoxon signed-rank test, ANOVA, or Kruskal-Wallis test was used to compare secondary outcomes, and blood chemistry and immunologic test results among groups. Duncan multiple range test or the Dwass-Steel-Critchlow-Fligner method was used for post-hoc analysis of statistically significant results.

## 3. Results

### 3.1. Patient Recruitment

Between March 2016 and December 2017, 91 patients participated in the screening test. Based on the inclusion/exclusion criteria, 11 patients were excluded from the trial. Twenty patients were assigned to each of the four treatment groups. ITT analysis was performed for 75 participants; patients who did not participate in the first validity evaluation after taking the medication (n = 5) were not included in this analysis. A total of 67 subjects completed the study. PP analysis was performed for 65 of these 67 participants; two patients whose compliance was <70% were excluded from this analysis. The entire study flow is shown in Figure 3.

### 3.2. Baseline Data

There were no significant baseline differences in terms of age, sex, height, weight, BMI, disease duration, and relevant history among the different groups (Table 3).

### 3.3. Primary Outcome

After 8 and 12 weeks of drug administration, there was no statistically significant difference in the degree of symptom improvement and symptom relief rate among the groups (Table 4). However, according to the CATMOD procedure (multiple logistic regression analysis) at week 12, SRS was more likely to alleviate symptoms than the placebo, and symptom improvement was found to be higher in males than in female subjects (*p* < 0.05, Table 5).

### 3.4. Secondary Outcomes

Secondary outcome results are summarized in Table 4. After eight weeks of drug administration, the SRS + OB and SRS groups were associated with significant improvements in abdominal discomfort when compared to that in the placebo group (*p* = 0.005). After 12 weeks, there were significant differences among the groups with respect to the following indicators: Abdominal pain (*p* = 0.030); abdominal discomfort (*p* < 0.001); frequency of abdominal pain (*p* = 0.005); BSF scale (*p* = 0.003). Base on post-hoc analysis, the SRS + OB group was associated with a statistically significant improvement compared to outcomes in the placebo group in terms of abdominal pain, abdominal discomfort, frequency of abdominal pain, and BSF scale. In addition, the patients in SRS + OB group showed significant improvements in abdominal discomfort, frequency of abdominal pain, and BSF scale when compared to symptoms in patients of the P-SRS + OB group. There were no statistically significant differences among the groups in terms of satisfaction of defecation, QOL, frequency of defecation, and force used for defecation.

### 3.5. Blood Chemistry and Iimmunologic Test Results

There were no significant differences among the groups in terms of blood chemistry and immunologic findings after eight weeks, as well as before the start of drug administration (Table 6).

### 3.6. Safety

There were three patient withdrawals, due to adverse reactions in the P-SRS + OB group (n = 1, ALT increase) and P-SRS + P-OB group (n = 2, abdominal pain or fever) (Figure 3). The reported information on adverse events is summarized in Table 7. There were no statistically significant differences among the groups with respect to each type of adverse reaction.

### 3.7. Compliance and Blinding Findings

As shown in Figure 3, there were two participants with compliance <70% in the P-SRS + OB (n = 1) and P-SRS + P-OB (n = 1) groups. No patients dropped out during the trial, due to blinding failure.

## 4. Discussion

This randomized clinical trial revealed that the co-administration of SRS and OB is potentially more effective in alleviating the severity of D-IBS (in terms of abdominal pain, abdominal discomfort, frequency of abdominal pain, BSF scale) than OB alone or placebo. Moreover, SRS monotherapy might be effective in improving D-IBS symptoms. Further, the SRS + OB group did not show any significant difference compare to the other groups in the terms of safety outcomes, including symptoms and laboratory tests. These results are consistent with our hypothesis that the co-administration of SRS and OB would be effective and safe for the treatment of D-IBS.

This trial had several strengths. First, this pilot study was a well-designed trial and, consequently, had a proper recruitment process and a low drop-out rate (~16%), generating high-quality data. Second, to our knowledge, this is the first clinical study approved by the KMFDS to investigate the efficacy and safety of the combination of a herbal drug and conventional treatment for D-IBS. Through the management of the institution, we were able to conduct the trial systematically. Third, although herbal medicines are generally difficult to standardize and manufacture, the SRS used in this study was produced with standard compounds according to K-GMP, and its composition was confirmed by HPLC-PDA.

However, it is difficult to conclude that this study provided sufficient evidence to support the co-administration of SRS and OB in clinical practice, because study results showed only limited improvements in D-IBS symptoms after co-administration. Although SRS, OB, and SRS + OB were more effective in improving the severity of various D-IBS symptoms than the placebo, the co-administration of SRS and OB was more effective than the single administration of each drug in only a few of the secondary outcome parameters. In addition, compared to that observed immediately after the end of dosing, more positive secondary outcome results were found four weeks after the end of treatment. Further, there were no significant differences in blood chemistry and immunologic test results, which means that SRS and OB alleviate IBS symptoms through mechanisms other than stress relief or immune response regulation.

We suggest that the limited outcomes were due to the inadequate control of several factors, in addition to the small sample size. First, we considered that the lactose used in SRS and the placebo might have affected the study outcome. We excluded patients with lactose intolerance and used a placebo that was prepared with an amount of lactose less than that known to cause lactose intolerance (400 mg/day) [44]. However, lactose is known to affect the digestive system of patients with D-IBS adversely; thus, a low dose of lactose might exacerbate symptoms [45]. In addition, we did not control and manage lactose intake in the patients’ daily lives, and the amount of lactose ingested was unknown.

Second, the study design did not consider any syndrome differentiation. Syndrome differentiation is a fundamental consideration for individual diagnosis, based on the oriental way of thinking, for traditional oriental drug treatment; for example, other medications might be considered depending on syndrome differentiation even for the same disease. In a previous study analyzing different RCTs involving the administration of a herbal drug to IBS patients, the proportion of patients with spleen-stomach weakness (proper syndrome differentiation type for SRS) was 49.4%, and studies that did not utilize syndrome differentiation showed poor treatment results [46]. Although SRS is widely used for abdominal pain and diarrhea, only limited effects can be expected without considering of syndrome differentiation. Therefore, in a further large-scale study, researchers should consider this factor for SRS administration, attempt to use lactose-free drugs, and implement a robust research design to collect information related to the patients’ daily diets [47].

Third, the reasons as to why the drugs resulted in better effects in terms of symptoms improvement at 12 weeks (follow-up period) compare to that at eight weeks (administration period) are unknown. This is because the laboratory tests conducted in this trial did not produce statistically significant results, but there was also no clear and objective indicator associated with the causes and symptoms of D-IBS. Therefore, we assume that the co-administration of SRS and OB had a positive effect on the intestinal environment and movement for D-IBS patients, but it is difficult to determine the underlying mechanism, as well as accurate period of drug administration for D-IBS symptom improvement. The treatment duration in this study followed clinical research guidelines related to IBS published in Korea, but it is debatable whether this period is sufficient to evaluate treatments for chronic diseases, such as IBS. According to a clinical study using OB for IBS, therapeutic benefits were more prominent than those with the placebo as of 4 to 10 weeks of administration [31,32]. Based on these previous studies and our results, future large-scale clinical studies might consider treatment periods longer than eight weeks.

Finally, pertinent information has been provided for the study design of future large-scale RCTs. As the formulation and dosing of SRS and OB used in the study were different, there might be advantages to using a double-dummy study design. However, if we focus on the fundamental purpose of the study, the design could be changed to a three-arm type, because the untreated placebo group was not included in the synergistic and safety assessments of the co-administration, and it could increase unnecessary enrollment efforts and ethical problems, due to the absence of proper treatment.

Interestingly, D-IBS symptom improvement was found to be better in males than in female subjects. In some previous studies, differences in IBS prevalence and symptoms according to sex were reported [3,48], but there have been no reports on differences in treatment outcomes. The mechanism underlying the effect of sex on D-IBS treatment remains to be elucidated. However, the sex-specific epidemiologic characteristics of IBS might have affected treatment outcomes. Further research on this will be needed, and future large-scale research should consider a sex-related research design.

In conclusion, despite several potential limitations as a pilot study, this trial shows several positive results and a systematic process to evaluate the safety and effectiveness of SRS and OB co-administration. In addition, this study indicated the feasibility of progressing to large-scale RCTs, to assess combinations of herbal and conventional drugs for D-IBS management.

## 5. Conclusions

The co-administration of SRS and OB was safe and was more effective in improving some symptoms (abdominal pain, discomfort, frequency of abdominal pain, BSF scale) of D-IBS than OB alone or placebo. Moreover, SRS monotherapy might be effective to improve D-IBS symptoms. In addition, this study was associated with an appropriate recruitment process and low drop-out and adverse events, and thus, could inspire future studies in these regards. However, because of limitations in the study design, such as low sample size, incompatibility of lactose used as a placebo, and the fact that syndrome differentiation was not considered for SRS administration, our results are not sufficient to conclude that the co-administration of SRS and OB is effective in relieving D-IBS symptoms. Nevertheless, we believe that these preliminary results might serve as a foundation for large-scale RCTs on combined SRS and OB intervention for D-IBS in the future.

## Figures and Tables

**Figure 1 jcm-08-01558-f001:**
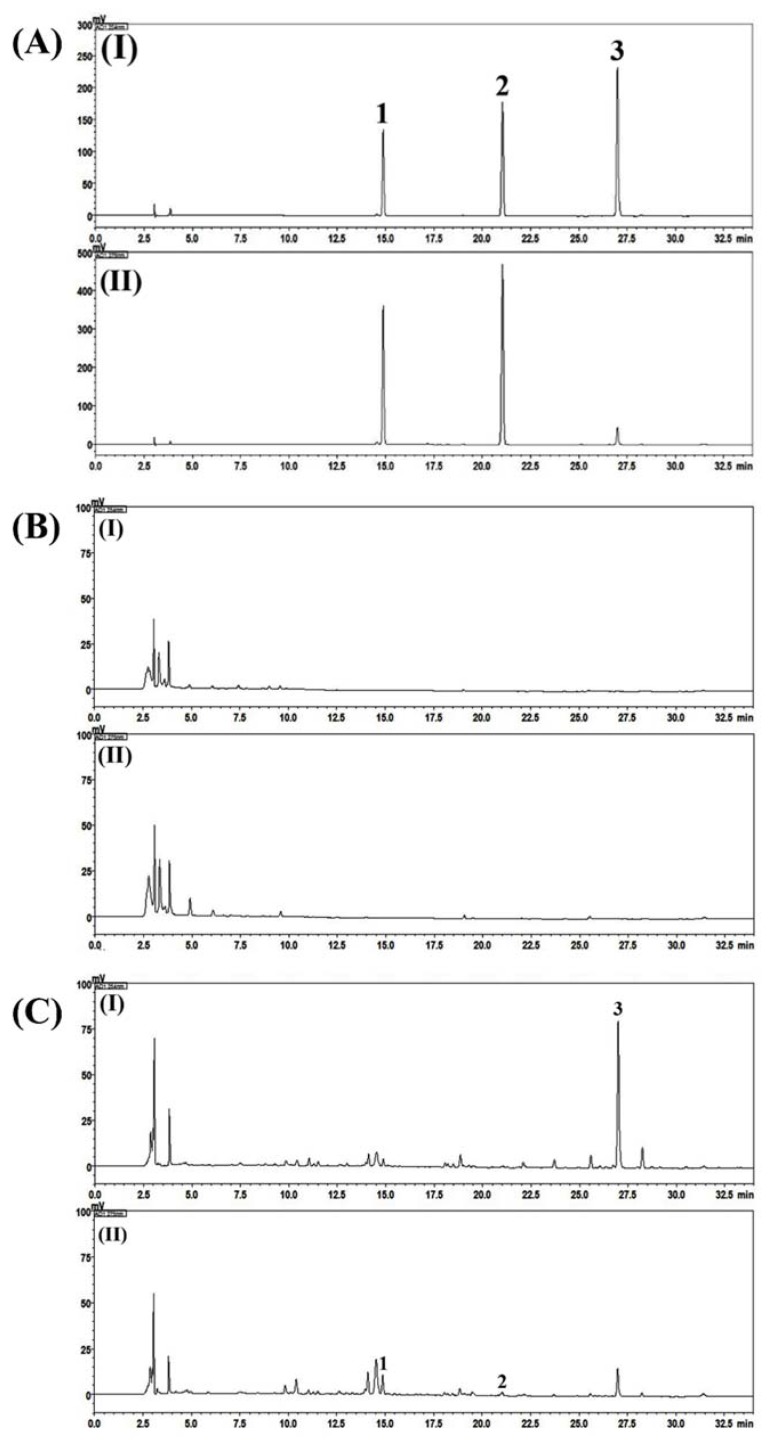
Representative HPLC-PDA chromatograms of the reference compounds (**A**), placebo (**B**) and Samryeongbaekchulsan sample (**C**) at 254 nm (**I**) and 275 nm (**II**). 1. Liquiritin, 2. liquiritigenin, 3. glycyrrhizin.

**Figure 2 jcm-08-01558-f002:**
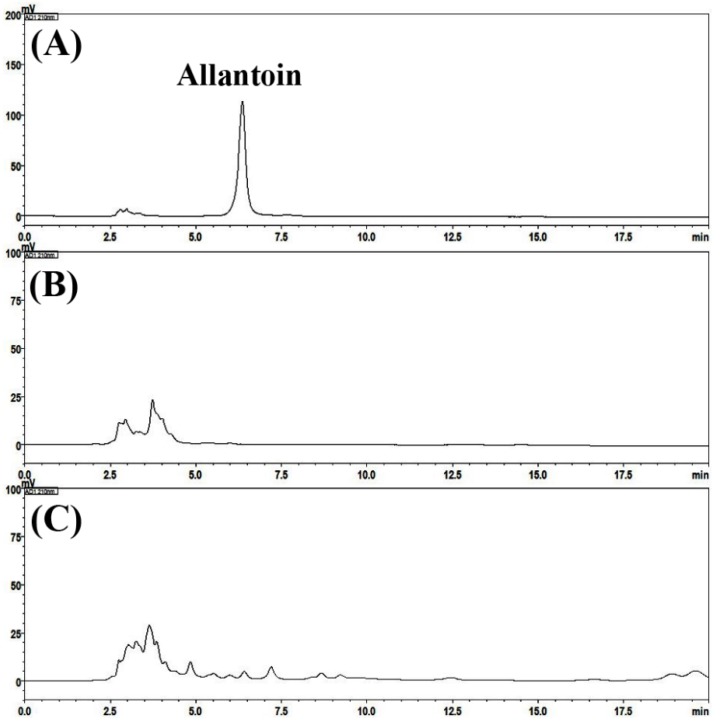
Representative HPLC-PDA chromatograms of the reference compound (**A**), placebo (**B**) and Samryeongbaekchulsan sample (**C**) at 210 nm.

**Figure 3 jcm-08-01558-f003:**
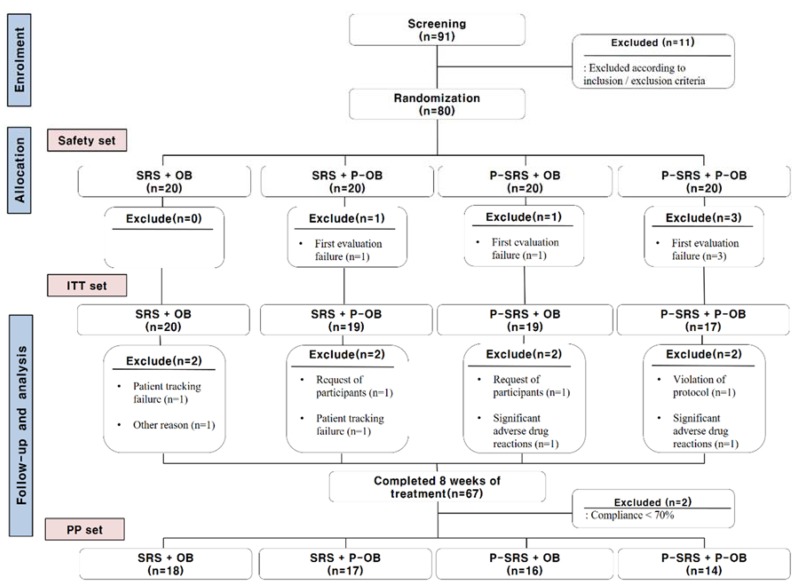
Flow chart of the study. SRS = Samryungbaekchul-san; P = placebo; OB = otilonium bromide; ITT = intent-to-treat; PP = per-protocol.

**Table 1 jcm-08-01558-t001:** Inclusion and exclusion criteria.

Inclusion Criteria	Exclusion Criteria
Aged 18–75 years Patients with the ability to read the symptom questionnaire and understand it, and agree with the clinical plan and voluntarily Patients satisfying the Rome III criteria of IBS Recurrent abdominal pain or discomfort that began at least six months before the clinical trial, with a rating of more than 3 on a 10-point scale, frequency of at least three days/month in the last three months and associated with two or more of the following (1) Improvement associated with defecation (2) Onset associated with a change in frequency of stools (3) Onset associated with a change in the form of stools D-IBS type is defined as loose/mushy or watery stools: Bristol Stool Form Scale 6–7 ≥ 2 times/week	Pregnancy, breast feeding Chronic liver disease (cirrhosis, chronic hepatitis B or C) Chronic renal failure or renal impairment (serum creatinine ≥ 2 times the upper limit of normal) Liver dysfunction (AST/ALT ≥ 3 times the upper limit of normal) Diabetes (HbA1c > 8% or not controlled by diet or medication) Hypertension (≥160/100 mmHg) Thyroid dysfunction (not controlled by proper medication) Significant hematologic, cardiac, pulmonary, neurological disorders, or other severe systemic disorders Abnormal findings on colonoscopy or colonography within the last five years History of surgery that affects gastrointestinal motility GI disease within six months before the clinical trial (inflammation of or ulcer in the esophagus, stomach, or duodenum; gastro-esophageal reflux disease; gastro-esophageal reflux disease; GI bleeding; GI stenosis or closure; infectious diarrhea; inflammatory bowel disease; pancreatic insufficiency; biliary abscess) Mental illness or an addiction to drugs or alcohol Use of medications that do not match the intent of this trial or have an interaction with clinical trial drugs, within two weeks before the trial Lactose intolerance (not controlled by food) Glaucoma Participation in other clinical studies within 30 days prior to clinical screening and have received other clinical trial medications Hypersensitivity to clinical trial medicines Other reasons; considered inappropriate for participation in clinical trials.

IBS = irritable bowel syndrome; D-IBS = diarrhea-predominant IBS; AST = aspartate aminotransferase; ALT = alanine aminotransferase; HbA1c = Hemoglobin A1c; GI = gastrointestinal.

**Table 2 jcm-08-01558-t002:** Ingredients of the herbal formula Samryungbaekchul-san.

Scientific Name	Plant Part Used	Dry Weight for Extraction (Gram/Pack)
Atractylodis Rhizoma Alba	Root	1.00
Poria Sclerotium	Dried core	1.33
Dioscoreae Rhizoma	Root	1.00
Glycyrrhizae Radix et Rhizoma	Root	0.50
Coicis Semen	Seed	2.67
Nelumbinis Semen	Seed	1.33
Platycodonis Radix	Root	0.83
Dolichoris Semen	Seed	1.33
Amomi Fructus	Fruit	0.67
Ginseng Radix	Root	1.00

**Table 3 jcm-08-01558-t003:** Baseline characteristics of the patients (Safety Set).

	SRS + OB (n = 20)	SRS + P-OB (n = 20)	P-SRS + OB (n = 20)	P-SRS + P-OB (n = 20)	*p*-Value
Age (year; mean; SD)	42.90 (15.13)	38.05 (15.27)	41.65 (14.26)	45.20 (13.56)	0.392 ^a^
Sex (n, %)					
Male	17 (85.00)	16 (80.00)	16 (80.00)	11 (55.00)	0.118 ^b^
Female	3 (15.00)	4 (20.00)	4 (20.00)	9 (45.00)	
Height (cm; mean; SD)	170.55 (7.39)	169.70 (5.96)	170.50 (5.12)	166.80 (9.42)	0.308 ^c^
Weight (Kg; mean; SD)	72.80 (11.54)	67.70 (13.79)	69.00 (8.99)	69.30 (15.02)	0.615 ^c^
BMI (kg/m^2^; mean; SD)	25.06 (3.97)	23.35 (3.75)	23.72 (2.62)	24.70 (3.55)	0.604 ^a^
Period of disease (year; mean; SD)	12.58 (10.59)	9.37 (7.05)	11.45 (9.16)	9.28 (11.57)	0.263 ^c^
History					
Surgical history (yes; n; %)	9 (45.00)	3 (15.00)	7 (35.00)	8 (40.00)	0.200 ^a^
Other diseases (yes; n; %)	13 (65.00)	7 (35.00)	9 (45.00)	11 (55.00)	0.262 ^a^
Prior medications (yes; n; %)	7 (35.00)	5 (25.00)	5 (25.00)	6 (30.00)	0.880 ^a^
CM (yes; n; %)	12 (60.00)	9 (45.00)	8 (40.00)	12 (60.00)	0.466 ^a^

^a^: Kruskal-Wallis test, ^b^: Chi-square test, ^c^: A one-way analysis of variance (ANOVA) test. SRS = Samryungbaekchul-san; OB = otilonium bromide; P = placebo; SD = standard deviation; BMI = body mass index; CM = concomitant medication use, which did not affect the trial.

**Table 4 jcm-08-01558-t004:** Outcome results (ITT analysis).

	SRS + OB (n = 20)	SRS + P-OB (n = 19)	P-SRS + OB (n = 19)	P-SRS + P-OB (n = 17)	*p*-Value
**Primary Outcome**:					
Degree of symptom improvement					
Week 8	1.67 ± 0.84	1.53 ± 0.62	1.76 ± 0.83	1.56 ± 0.51	0.867 *
Week 12	1.61 ± 0.61	1.59 ± 0.71	1.94 ± 0.66	1.87 ± 0.64	0.265 *
Symptoms relief rate(improved, n, %)					
Week 8	8 (44.44)	9 (52.94)	8 (47.06)	7 (43.75)	0.949 ^†^
Week 12	8 (44.44)	9 (52.94)	4 (23.53)	4 (26.67)	0.233 ^†^
**Secondary Outcomes** (**Severity of D-IBS Symptoms):**					
Abdominal Pain					
Week 8	−2.89 ± 1.84	−2.71 ± 2.34	−2.00 ± 2.60	−1.94 ± 1.57	0.454 ^‡^
Week 12	−2.94 ± 2.34 ^a^	−2.88 ± 2.39 ^a^	−1.53 ± 1.50 ^a,b^	−1.13 ± 2.07 ^b^	0.030 ^‡^
Abdominal Discomfort					
Week 8	−3.72 ± 1.87 ^a^	−3.00 ± 1.54 ^a^	−2.53 ± 2.35 ^a,b^	−1.25 ± 2.02 ^b^	0.005 ^‡^
Week 12	−3.67 ± 2.11 ^a^	−3.76 ± 1.82 ^a^	−2.35 ± 1.66 ^b^	−1.00 ± 1.85 ^c^	< 0.001 ^‡^
Satisfaction of Defecation					
Week 8	−2.67 ± 3.05	−2.94 ± 1.71	−2.71 ± 2.69	−3.25 ± 2.32	0.954 *
Week 12	−3.33 ± 2.47	−3.65 ± 2.00	−2.47 ± 2.50	−2.40 ± 2.13	0.310 ^‡^
Quality of Life					
Week 8	−3.17 ± 2.28	−2.71 ± 2.37	−2.65 ± 3.06	−2.44 ± 2.03	0.941 *
Week 12	−3.17 ± 2.18	−3.47 ± 2.70	−2.35 ± 2.29	−1.80 ± 2.43	0.325 *
Frequency of Abdominal Pain					
Week 8	−3.33 ± 2.11	−3.00 ± 1.84	−2.41 ± 2.79	−2.38 ± 2.25	0.539 ^‡^
Week 12	−3.56 ± 2.01 ^d^	−3.24 ± 2.46 ^d, e^	−1.82 ± 1.42 ^e^	−1.33 ± 2.44 ^e^	0.005 *
Frequency of Defecation					
Week 8	−0.86 ± 1.23	−0.76 ± 0.90	−0.53 ± 0.70	−.84 ± 0.85	0.723 ^‡^
Week 12	−0.89 ± 1.22	−0.85 ± 1.20	−0.59 ± 0.69	−0.80 ± 1.01	0.706 *
Bristol Stool Form Scale					
Week 8	−0.94 ± 1.06	−1.35 ± 0.79	−1.24 ± 0.90	−1.44 ± 0.51	0.185 *
Week 12	−1.33 ± 0.59 ^d^	−1.41 ± 0.94 ^d^	−0.65 ± 0.61 ^d^	−0.80 ± 0.68 ^d, e^	0.003 *
Force used for Defecation					
Week 8	−0.06 ± 0.24	0.00 ± 0.00	0.00 ± 0.35	−0.06 ± 0.25	0.803 *
Week 12	−0.06 ± 0.24	−0.00 ± 0.00	−0.06 ± 0.24	−0.07 ± 0.26	0.783 *

1. Values are mean ± standard deviation (SD), except symptoms relief rate. 2. The results reported as negative numbers indicate a decrease from the baseline. 3. The same letters indicate non-significant difference between the groups: (1) ^a,b,c^: Duncan multiple range test; (2) ^d,e^: Dwass-Steel-Critchlow-Fligner Method. 4. The statistical methods according to each symbol were used as follows: (1) *: Kruskal-Wallis test; (2) †: Chi-square test; (3) ‡: A one-way analysis of variance (ANOVA) test. ITT = intent-to-treat; SRS = Samryungbaekchul-san; OB = otilonium bromide; P = placebo; D-IBS = diarrhea-predominant irritable bowel syndrome.

**Table 5 jcm-08-01558-t005:** Multiple logistic regression analysis of risk factors for symptom relief (ITT analysis).

Variables	OR (95% CI)	*p*-Value
Intercept Group (Placebo)		0.746
SRS + OB	4.84 (0.85–27.74)	0.077
OB	1.15 (0.20–6.64)	0.880
SRS	5.47 (1.01–29.53)	0.049
Sex (Male)	0.21 (0.05–0.82)	0.025
Other Disease (Yes)	0.26 (0.06–1.11)	0.068

ITT = intent-to-treat; OR = odds ratio; CI = confidence interval; SRS = Samryungbaekchul-san; OB = otilonium bromide.

**Table 6 jcm-08-01558-t006:** Blood chemistry and immunologic test results (ITT analysis).

	SRS + OB (n = 20)	SRS + P-OB (n = 19)	P-SRS + OB (n = 19)	P-SRS + P-OB (n = 17)	*p*-Value
Cortisol (ng/mL)	4.45 ± 25.83	−10.38 ± 43.58	−8.70 ± 51.41	−2.48 ± 27.70	0.392 *
CRH (pg/mL)	−6.11 ± 14.77	3.06 ± 13.71	−0.58 ± 11.38	−0.05 ± 11.09	0.469 *
Serotonin ng/mL)	−3.66 ± 47.77	−32.62 ± 78.77	−14.99 ± 54.01	2.28 ± 65.21	0.470 ^†^
Eotaxin (pg/mL)	−8.13 ± 19.34	−5.61 ± 7.71	5.25 ± 23.35	−5.92 ± 17.98	0.093 *
FGF basic (pg/mL)	−2.30 ± 5.30	−2.38 ± 5.56	0.30 ± 6.68	0.73 ± 3.24	0.301 *
G-CSF (pg/mL)	−0.97 ± 2.70	−1.30 ± 1.96	−0.51 ± 4.97	0.68 ± 3.61	0.445 *
GM-CSF (pg/mL)	−0.51 ± 4.27	−2.64 ± 2.92	−1.84 ± 3.97	0.48 ± 2.17	0.193 *
IFN-gamma (pg/mL)	4.53 ± 16.77	−2.38 ± 7.31	−6.69 ± 15.11	1.07 ± 8.40	0.587 *
IL-1 beta (pg/mL)	0.00 ± 0.86	−0.10 ± 0.27	0.14 ± 0.60	0.00 ± 0.23	0.640 *
IL-1 RA (pg/mL)	−1.71 ± 11.86	4.19 ± 17.80	−6.71 ± 13.56	2.30 ± 9.51	0.718 *
IL-2 (pg/mL)	−0.21 ± 1.24	−0.42 ± 0.00	−3.07 ± 0.98	ND	0.172 *
IL-4 (pg/mL)	0.19 ± 0.91	−0.04 ± 0.33	−0.23 ± 0.73	0.04 ± 0.31	0.481 *
IL-5 (pg/mL)	−0.64 ± 0.25	−0.16 ± 2.06	0.71 ± 3.56	0.76 ± 2.60	0.903 ^†^
IL-6 (pg/mL)	−0.06 ± 1.21	0.03 ± 0.62	−0.02 ± 1.09	−0.15 ± 0.27	0.771 *
IL-7 (pg/mL)	−0.32 ± 1.64	0.19 ± 1.76	−0.32 ± 1.08	0.06 ± 1.00	0.878 *
IL-8 (pg/mL)	0.68 ± 3.01	−1.23 ± 3.19	−1.66 ± 2.93	2.37 ± 4.78	0.123 *
IL-9 (pg/mL)	−2.71 ± 12.25	−1.41 ± 2.70	−3.35 ± 8.75	1.16 ± 7.56	0.552 *
IL-10 (pg/mL)	−0.24 ± 0.90	−0.16 ± 1.17	−2.74 ± 6.72	−0.13 ± 0.38	0.480 *
IL-12 (p70) (pg/mL)	−1.09 ± 2.90	−3.00 ± 6.44	2.12 ± 8.80	0.63 ± 5.26	0.538 *
IL-13 (pg/mL)	0.11 ± 2.66	−0.56 ± 1.06	−0.82 ± 2.98	0.65 ± 1.84	0.500 ^†^
IL-15 (pg/mL)	ND	ND	ND	ND	ND
IL-17 (pg/mL)	−0.24 ± 2.67	−2.55 ± 4.16	−1.77 ± 6.37	−0.26 ± 6.77	0.607 ^†^
IP-10 (pg/mL)	1.23 ± 178.88	−55.64 ± 212.22	−78.88 ± 170.02	63.85 ± 247.66	0.428 *
MCP-1 (pg/mL)	0.14 ± 7.75	−1.12 ± 5.79	−0.77 ± 12.52	0.53 ± 3.86	0.856 *
MIP-1a (pg/mL)	0.03 ± 0.36	−0.23 ± 0.51	−0.10 ± 0.42	0.06 ± 0.23	0.202 *
MIP-1b (pg/mL)	−2.96 ± 13.98	−1.66 ± 6.85	−5.11 ± 13.40	3.41 ± 11.04	0.448 *
PDGF-bb (pg/mL)	−226.94 ± 524.07	18.55 ± 599.49	−138.69 ± 751.16	86.78 ± 500.26	0.758 *
RANTES (pg/mL)	64.52 ± 549.89	88.36 ± 362.78	−225.42 ± 410.71	101.05 ± 558.43	0.215 *
TNF-alpha (pg/mL)	−8.88 ± 29.72	−1.23 ± 3.67	7.12 ± 41.34	−1.96 ± 7.29	0.904 *
VEGF (pg/mL)	−3.92 ± 12.41	−2.18 ± 15.99	−1.36 ± 17.70	−2.58 ± 15.81	0.998 *

1. Values are mean ± standard deviation (SD). 2. The results reported as negative numbers indicate a decrease from the baseline. 3. The statistical methods according to each symbol were used as follows: (1) *: Kruskal-Wallis test; (2) †: A one-way analysis of variance (ANOVA) test. ITT = intent-to-treat; SRS = Samryungbaekchul-san; OB = otilonium bromide; P = placebo; D-IBS = diarrhea-predominant irritable bowel syndrome; CRH = corticotropin-releasing hormone; EFG = fibroblast growth factor; G = granulocyte; CSF = colony stimulating factor; GM = granulocyte-macrophage; IFN = interferon; IL = interleukin; RA = receptor antagonist; ND = not detected; IP = interferon gamma-induced protein; MCP = human monocyte chemoattractant protein; MIP = macrophage inflammatory proteins; PDGF = platelet-derived growth factor; RANTES = regulated on activation, normal T cell expressed and secreted; TNF = tumor necrosis factor; VEGF = vascular endothelial growth factor.

**Table 7 jcm-08-01558-t007:** Summarized results of safety test (safety set).

	SRS + OB (n = 20)	SRS + P-OB (n = 20)	P-SRS + OB (n = 20)	P-SRS + P-OB (n = 20)	*p*-Value
AE (n; %)	6 (30.00)	7 (35.00)	5 (25.00)	8 (40.00)	0.778 ^a^
ADR (n; %)	1 (5.00)	0 (0.00)	0 (0.00)	1 (5.00)	1.000 ^b^
SAE (n; %)	0 (0.00)	0 (0.00)	1 (5.00)	2 (10.00)	0.610 ^b^
SADE (n; %)	0 (0.00)	0 (0.00)	0 (0.00)	1 (5.00)	1.000 ^b^

^a^: Chi-square test, ^b^: Fisher’s exact test. SRS = Samryungbaekchul-san; OB = otilonium bromide; P = placebo. AE = adverse events; ADR = adverse drug reaction; SAE = serious adverse event; SADE = serious adverse drug event.

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
