# Peer review of "Effect of Samryungbaekchul-san Combined with Otilonium Bromide on Diarrhea-Predominant Irritable Bowel Syndrome: A Pilot Randomized Controlled Trial"

_jcm, 2019, doi:10.3390/jcm8101558_

Round 1

Reviewer 1 Report

The present manuscript is potentially interesting and, correctly, the authors recognised several limits of their research. However, there are some theoretical and methodological points that need to be accurately revised and eventually integrated. Following are notes and suggestions to improve the manuscript.

Main concerns:

I do not understand the rationale of the study: the patients were treated for 8 weeks but the measurements were made after 12 weeks, in other words after 4 weeks of drugs withdrawal. Why did the authors wait for this period of abstinence? Why did not they make the evaluations also after 8 weeks. The importance of having beneficial effects of the treatments after the withdrawal is matter of debate especially for those drugs (i.e. OB) that are considered mainly symptomatic. (see the last paper by Clavè and coworkers.

I do have doubts on the length of the treatment (8 weeks) that it is quite short.

In the results:

The sum of the numbers in the last row of Figure 3 is 65 and not 67 as reported in the text. The number of patients reported in Table 4 and 5 is 75 thus, also the patients that did not complete for several reasons the protocol are included. I wonder whether this is correct for getting statistically (and clinically) significant information. In table 5 no sex reference is evident

Finally, the findings reported suggest that only SRS is beneficial to the patients and the co-treatment besides the lack of adverse effects, does not seem to change significantly the outcomes. Is this correct? Please discuss this point.

In the introduction the studies using the co-treatment with SRS and OB have to be listed in the references.

In the methods the time table of the protocol should be adequately reported (a short information is present only in the abstract, rows 27-28)

Minor points:

Make order in the quotation of the Rome Criteria (III or IV)

What the authors mean with 2 weeks of preparation?

In the introduction an updating review on IBS by Vannucchi and Evangelista published in the Journal of Clinical Medicine in 2018, should be mentioned

Table 6: SAE not SAR

Discussion row 252: These, not theses.

Author Response

Response to Reviewer 1.

Comments and Suggestions for Authors

The present manuscript is potentially interesting and, correctly, the authors recognised several limits of their research. However, there are some theoretical and methodological points that need to be accurately revised and eventually integrated. Following are notes and suggestions to improve the manuscript.

We thank the reviewer for the suggestions, which have greatly helped us in improving the manuscript. We agree with the reviewer’s comments and have incorporated the suggestions in the revised manuscript. We hope that the revisions meet the reviewer’s expectations.

Main concerns:

Q) I do not understand the rationale of the study: the patients were treated for 8 weeks but the measurements were made after 12 weeks, in other words after 4 weeks of drugs withdrawal. Why did the authors wait for this period of abstinence? Why did not they make the evaluations also after 8 weeks. The importance of having beneficial effects of the treatments after the withdrawal is matter of debate especially for those drugs (i.e. OB) that are considered mainly symptomatic. (see the last paper by Clavè and coworkers.

A) We thank the reviewer for this comment. We have also assessed the patients at 8 weeks after dosing, as indicated in ‘2.5 Study Outcomes’, and the results have been added in a new Table 4. We have provided an interpretation of these additional results in the Discussion section, in conjunction with the need to improve trial design of future large-scale studies.

Q) I do have doubts on the length of the treatment (8 weeks) that it is quite short.

A) We agree with the reviewer. However, according to the “Guidelines for clinical trials of irritable bowel syndrome therapeutics,” published by the National Institute of Food and Drug Safety Evaluation in Korea (December, 2015), clinical studies on IBS are only required to have an administration period of more than 8 weeks. Although the 8-week period may be short, we think this period does not deviate significantly from the clinical guidelines and the trial periods in previous studies. This was a pilot study; however, we will design a future large-scale clinical study with a longer administration period. In addition, based this comment, we have detailed the limited treatment period as a study limitation in the Discussion section.

Q) In the results:

The sum of the numbers in the last row of Figure 3 is 65 and not 67 as reported in the text. The number of patients reported in Table 4 and 5 is 75 thus, also the patients that did not complete for several reasons the protocol are included. I wonder whether this is correct for getting statistically (and clinically) significant information.

A) We thank the reviewer for carefully reviewing our paper. As mentioned in the manuscript, 67 patients participated in this study, with all the schedules completed; however, two patients were excluded from the per-protocol (PP) analysis because of <70% compliance. To the best of our knowledge, the intent-to-treat (ITT) analysis for evaluating effectiveness must consider only patients who were followed up at least once after randomization, regardless of participation until the end of the study (n = 75). Therefore, the number of patients included in Tables 4 and 5 for the ITT analysis was 75. We hope this answer is satisfactory.

Q) In table 5 no sex reference is evident.

A) This information was indeed missing in Table 5; we are sorry for having overlooked this. We have added new data on variables (sex and other diseases) in Table 5. In addition, we have included information on possible reasons for gender-related outcomes in the Discussion section.

Q) Finally, the findings reported suggest that only SRS is beneficial to the patients and the co-treatment besides the lack of adverse effects, does not seem to change significantly the outcomes. Is this correct? Please discuss this point.

A) We thank the reviewer for this apt question. As pointed out by the reviewer, an additional effect of co-administration was not observed in the primary outcome, but it did improve the severity of D-IBS compared to OB alone or placebo in terms of abdominal pain, abdominal discomfort, frequency of abdominal pain, and BSF scale in the secondary outcome parameters. Although these results may be limited, we think they suggest new possibilities for potential effectiveness of co-administration. We hope to provide more convincing results in future large-scale studies.

Q) In the introduction the studies using the co-treatment with SRS and OB have to be listed in the references.

A) We thank the reviewer for this suggestion. However, there are very few clinical studies related to co-administration of herbal and conventional medicine in D-IBS. In addition, as there are no clinical studies in which SRS and OB were administered together, it is impossible to cite previous studies. Therefore, we have provided sufficient clinical research findings in the Introduction for each of SRS and OB.

Q) In the methods the time table of the protocol should be adequately reported (a short information is present only in the abstract, rows 27-28)

A) We thank the reviewer for this advice. Although not in the form of a table, we have described details on the study progression schedule in 2.2 Study Design.

Minor points:

Q) Make order in the quotation of the Rome Criteria (III or IV)

A) In 2015, when the study was planned and designed (this has been clarified in the text now), the most recent ROME criteria for validity in Korea were the ROME III criteria. Therefore, this trial was planned and conducted based on the ROME III criteria. However, before submitting this article, ROME IV criteria were published, and therefore, we referred to ROME IV criteria in the Introduction, independently of our trial. We hope that this answer clarifies the reference to both versions of these criteria.

Q) What the authors mean with 2 weeks of preparation?

A) The preparation period was a “run-in period,” during which patients were screened for eligibility to partake in the clinical study. We have explained this in 2.2 Study Design.

Q) In the introduction an updating review on IBS by Vannucchi and Evangelista published in the Journal of Clinical Medicine in 2018, should be mentioned

A) We thank the reviewer for this excellent recommendation. This paper is indeed very interesting and enriches our article. We have referred to this paper in the revised manuscript, and we have added more up-to-date information on IBS in the Introduction.

Q) Table 6: SAE not SAR, Discussion row 252: These, not theses.

A) We thank the reviewer for pointing out these typos; we have corrected them accordingly.

Reviewer 2 Report

Congratulations, a well-designed trial and, consequently, it generated high-quality data, proper recruitment process.

Some requests to the authors:

In table 3, regarding the P-SRS + P-OB the sum of male (11) and female (20) is different from the number of the total patients (20 patients) Give more data about: the meaning of “other history”, what “concomitant medications” are took by the patients Give more explanation about what “Syndrome differentiation” is for the non-oriental readers

Author Response

Response to Reviewer 2.

Comments and Suggestions for Authors

Congratulations, a well-designed trial and, consequently, it generated high-quality data, proper recruitment process.

We thank the reviewer for this positive evaluation of our work.

Some requests to the authors:

Q) In table 3, regarding the P-SRS + P-OB the sum of male (11) and female (20) is different from the number of the total patients (20 patients)

A) We thank the reviewer for careful review of our study. We have corrected the number of female subjects (n = 9).

Q) Give more data about: the meaning of “other history”, what “concomitant medications” are took by the patients Give more explanation about what “Syndrome differentiation” is for the non-oriental readers

A) We thank the reviewer for this helpful comment. We have rephrased “other history” as “other disease,” and we have clarified “concomitant medications” in more detail in a footnote to Table 3. We have also explained in detail what “syndrome differentiation” is an individual diagnostic method based on oriental medical concepts.

Reviewer 3 Report

In the paper entitled: Effect of Samryungbaekchul-san combined with otilonium bromide on diarrhea- predominant irritable bowel syndrome: a pilot randomized controlled trial” (manuscript no. jcm-572127) the Authors describe preliminary data to assess the usefulness of co-administration of herbal extract from Samryungbaekchul-san (SRS) and conventional drug (otilonium bromide- OB) in relieving D-IBS symptoms, including diarrhea-like stool patterns and abdominal discomfort or pain. They have found that co-administration of herbal and conventional drugs was effective in improving severity of D-IBS, measured by abdominal pain, abdominal discomfort, frequency of abdominal pain, and Bristol stool form scale, compared to OB alone or placebo use. The combination of the herbal drug and conventional treatment did not lead to significant changes in the laboratory blood tests, including cortisol, CRH, serotonin, as well as cytokines and growth factors,  and can be safely used in clinical practice for management of D-IBD syndrome. The manuscript shows good preparation of the Authors for taken theme. The references are chosen properly for topic of this work. Individual positions are cited correctly in the text. Despite the many limitations, most of which were mentioned in the Discussion section, conception of this study is interesting and manuscript makes some contribution to the present knowledge about the efficacy and safety of the co-administration of conventional and traditional herbal drugs in D-IBS treatment.

Detailed comments are presented below.

More detailed information concerning the clinical justification for the choice of chemical and immunological parameters assessed in the blood, should be added into the text. Figure 1 and 2 are very poor quality and difficult to read. Because they can be very helpful to the reader, the Authors should replace these with the better quality photographs. In the Results section (3. Primary outcome subsection) the Authors state that “……,SRS was more likely to alleviate symptoms than the placebo, and the improvement in symptoms was found to be higher in males than in females (p<0.05, Table 5)”. I can't find gender-related differences in Table 5. Moreover, it would be interesting to discuss the possible reasons for the observed sex-related changes. This issue is completely omitted in the  Discussion section.

Author Response

Response to Reviewer 3.

Comments and Suggestions for Authors

In the paper entitled: “Effect of Samryungbaekchul-san combined with otilonium bromide on diarrhea- predominant irritable bowel syndrome: a pilot randomized controlled trial” (manuscript no. jcm-572127) the Authors describe preliminary data to assess the usefulness of co-administration of herbal extract from Samryungbaekchul-san (SRS) and conventional drug (otilonium bromide- OB) in relieving D-IBS symptoms, including diarrhea-like stool patterns and abdominal discomfort or pain. They have found that co-administration of herbal and conventional drugs was effective in improving severity of D-IBS, measured by abdominal pain, abdominal discomfort, frequency of abdominal pain, and Bristol stool form scale, compared to OB alone or placebo use. The combination of the herbal drug and conventional treatment did not lead to significant changes in the laboratory blood tests, including cortisol, CRH, serotonin, as well as cytokines and growth factors, and can be safely used in clinical practice for management of D-IBD syndrome. The manuscript shows good preparation of the Authors for taken theme. The references are chosen properly for topic of this work. Individual positions are cited correctly in the text. Despite the many limitations, most of which were mentioned in the Discussion section, conception of this study is interesting and manuscript makes some contribution to the present knowledge about the efficacy and safety of the co-administration of conventional and traditional herbal drugs in D-IBS treatment.

We thank the reviewer for careful review of our paper and this positive evaluation of our work.

Detailed comments are presented below.

Q) More detailed information concerning the clinical justification for the choice of chemical and immunological parameters assessed in the blood, should be added into the text.

A) We thank the reviewer for thoughtful review. We have added a clinical justification for the choice of parameters in 2.5.3 Blood Chemistry and Immunologic Tests, as suggested.

Q) Figure 1 and 2 are very poor quality and difficult to read. Because they can be very helpful to the reader, the Authors should replace these with the better quality photographs.

A) In agreement with this comment, we have provided larger images and increased image resolution to 300 dpi.

Q) In the Results section (3. Primary outcome subsection) the Authors state that “……,SRS was more likely to alleviate symptoms than the placebo, and the improvement in symptoms was found to be higher in males than in females (p<0.05, Table 5)”. I can't find gender-related differences in Table 5. Moreover, it would be interesting to discuss the possible reasons for the observed sex-related changes. This issue is completely omitted in the Discussion section.

A) We thank the reviewer for this thoughtful comment. As pointed out by the reviewer, this information was missing in Table 5; the relevant data have been added. In addition, as advised, we have discussed possible reasons for the gender-related outcome in the Discussion section. However, the exact reasons for and mechanisms underlying this gender-dependent difference in outcome remains to be studied in future studies.

Round 2

Reviewer 1 Report

I appreciate the efforts the authors made in improving the text and answering to my comments, notwithstanding, the authors' answers did not satisfy all my requests and concerns. in particular:

- they add the data at the 8 weeks but did not adequately discuss the meaning of the results (8 ws 12 weeks)

-they did not convince me that OB has any effects that could justify the double treatment

- they should remove the sentence that affirm that few studies have been done on this kind of double treatment if it has never done before (see abstract and introduction and discussion)

- they did not explain the diverse number of patients reported in the text respect to the sum obtained in figure 3.

Finally,  I found several grammar and syntax mistakes.

Author Response

# Response to Reviewer 1

 I appreciate the efforts the authors made in improving the text and answering to my comments, notwithstanding, the authors' answers did not satisfy all my requests and concerns. in particular:

We would like to thank you for carefully evaluating our manuscript. Based on your suggestions, we believe that the manuscript has been markedly improved. We hope that the manuscript is now suitable for publication. Please find with our resubmission a point-by-point response to your concerns.

Q) They add the data at the 8 weeks but did not adequately discuss the meaning of the results (8 ws 12 weeks)

A) We thank the reviewer for this helpful comment. Based on your advice, we have provided a more in-depth interpretation of the additional results and the extension of the medication period for future large scale studies in the Discussion section. We hope that this answer is satisfactory.

Q) They did not convince me that OB has any effects that could justify the double treatment

A) We thank the reviewer for pointing this out. At the time of study planning, we found that the main effects of SRS (improvement of diarrhea) and OB (improvements of abdominal discomfort and abdominal pain) were different. Accordingly, we hypothesized that combined treatment could result in synergistic effects based on the advantages of each drug. We describe this in more detail in the Introduction section. Although limited, our study also found that OB was more effective than the placebo and that SRS + OB was superior to OB at weeks 12 with respect to the alleviation of abdominal discomfort. This is an example showing the justification of the combined treatment based on OB.

Q) They should remove the sentence that affirm that few studies have been done on this kind of double treatment if it has never done before (see abstract and introduction and discussion)

A) We have followed your advice and removed the term 'few studies', mentioned in the Abstract and Introduction. We have also revised these sentences accordingly.

Q) They did not explain the diverse number of patients reported in the text respect to the sum obtained in figure 3.

A) We thank the reviewer for carefully reviewing our paper and pointing this out. We modified Figure 3 based on your advice to make it easier to understand the number of patients described in the text. Although there was no direct modification of the text, we think it will be easier for the reader to understand these diverse numbers.

Q) Finally, I found several grammar and syntax mistakes.

A) In accordance with this comment, we have enlisted the help of native English speakers to perform additional grammar and syntax corrections throughout this manuscript.